# Correlations between Problematic Mobile Phone Use and Depressiveness and Daytime Sleepiness, as Well as Perceived Social Support in Adolescents

**DOI:** 10.3390/ijerph192013549

**Published:** 2022-10-19

**Authors:** Marta Nowak, Kamila Rachubińska, Małgorzata Starczewska, Ewa Kupcewicz, Aleksandra Szylińska, Aneta Cymbaluk-Płoska, Elżbieta Grochans

**Affiliations:** 1Department of Nursing, Faculty of Health Sciences, Pomeranian Medical University, Żołnierska 48, 71-210 Szczecin, Poland; 2Department of Nursing, Collegium Medicum, University of Warmia and Mazury in Olsztyn, 14 C Zolnierska Street, 10-719 Olsztyn, Poland; 3Department of Medical Rehabilitation and Clinical Physiotherapy, Pomeranian Medical University, Żołnierska 54, 71-210 Szczecin, Poland; 4Department of Gynecological Surgery and Gynecological Oncology of Adults and Adolescents, Pomeranian Medical University, 70-111 Szczecin, Poland

**Keywords:** smartphone addiction, addiction, behavioral addiction, young adults

## Abstract

(1) The aim of this study was to estimate the scale of mobile phone addiction among young adults as well as to establish whether the low level of perceived social support is related to problematic smartphone use, and whether an addictive pattern of mobile phone use is related to the prevalence of depressiveness and excessive daytime sleepiness. (2) The study was carried out using the diagnostic poll method via the questionnaire technique. Both the author’s own questionnaire and the following standardized research tools were used: the Mobile Phone Problem Use Scale for Adolescents (MPPUSA), the Beck Depression Inventory (BDI), the Epworth Sleepiness Scale (ESS), and the Multidimensional Scale of Perceived Social Support (MSPSS). (3) Perceived social support was significantly lower in the group of respondents who problematically used their mobile phones in comparison with the ones who used them in a proper way. Severity of depressive symptoms and daytime sleepiness was significantly higher in respondents addicted to their mobiles compared to nonaddicted ones. (4) Conclusions: An important correlation between phone addiction and the prevalence of depressive symptoms and excessive daytime sleepiness exists. Problematic mobile phone use concerns individuals with low levels of perceived social support.

## 1. Introduction

This paper is organized into six sections. Section 1 presents the theoretical foundation of the studied phenomenon of smartphone addiction. Section 2 presents the study design, the methods used, and the description of the statistical analysis. The description of the study results and their graphical illustration in the form of tables are presented in Section 3. Section 4 discusses the results of the study, which were compared with the results of different researchers. The limitations of the study are presented in Section 5, while conclusions and proposals for future work are discussed in Section 6.

The first invented telephone, patented in 1986, differed radically not only in appearance, but above all in the multitude of functions offered in relation to currently available smartphones. Even comparing the first prototype of a mobile phone from the 1960s to the latest models, one can observe a huge technological progress and the expansion of the possibilities offered, which integrally permeate human functioning in many aspects of everyday life. Basic applications such as the ability to make calls and write text messages, combined with the advanced capabilities offered by applications (social media, health management, finances) made the number of smartphone users in 2016 (3.7 billion) almost double in 2021 (6.3 billion), and according to forecasts, in the next five years this number will increase by several hundred million [1].

In recent decades, the increased use of a mobile phone has become associated with problematic mobile phone use (PMPU), which translates into worse functioning in terms of somatic and mental health [2], as well as a reduced quality of life level (QOL) and health-related quality of life (HRQOL) [3,4,5].

The phenomenon of addiction to a mobile phone, although classified as a behavioral addiction, has an unspecified nosological position and has no clear definition. The 2013 Diagnostic and Statistical Manual of Mental Disorders DSM-5 identified gambling disorder as the only behavioral addiction in the non-substance-use disorder category. Moreover, in the International Statistical Classification of Diseases and Health Problems ICD-11, in the category of mental, behavioral, or neurodevelopmental disorders and the subcategory of disorders resulting from addictive behavior, only gambling disorder (both online and offline) was distinguished. Problematic mobile phone use has been defined, partly, as a potential behavioral addiction through the use of mobile phones or smartphones. PMPU can cause issues for users when enacted as an excessive form of relationship maintenance or in banned or dangerous situations during addictive use [6,7].

Significant symptoms of PMPU include nomophobia, phantom phone signals, and phubbing. Nomophobia is an abbreviation of a phrase “no mobile phone” and is defined as the fear of being detached from one’s mobile phone. The phenomenon manifests as a feeling of discomfort, anxiety, nervousness and fear, frustration, or even aggression, depending on the level of risk of the inability to use the phone, or on the duration of the separation [8,9]. Phantom phone signals refer to phantom vibration (PV) and phantom ringing (PR) of the mobile phone. The essence of the phenomenon lies in experiencing individual vibrations, hearing their phone ring or seeing the phone light up when it does not actually take place [10]. The term phubbing is a neologism resulting from the combination of the words phone and snubbing and refers to the disregard of the interlocutor in the social environment for the benefit of using a mobile phone. It has been observed that phubbing behavior is a common phenomenon among adolescents (49.3%) and is closely related to both the problematic use of the Internet and phones [11,12,13].

### 1.1. Controversy around a Behavioral Addiction

Opponents of the phrase “behavioral addiction” believe that it is stigmatizing and pathologizing behavior within the normal limits, which is a natural result of continuous technological progress. Researchers also point out that the analysis and definition of behavioral addictions should take into account not only the sociocultural context, but also the professional or university context, so as to avoid misinterpretation of data. As an example of the importance of sociocultural determinants, we can mention Asian countries, whose moral standards translate into increased use of a mobile phone. A critical approach is also shown in connection with the permanent supplementation of the group of behavioral addictions with newer and newer phenomena, which in consequence are incomparable with other behavioral addictions, and even more so with substance addictions [14,15].

According to the opponents, such a multidirectional expansion of behavioral addictions, we can also put the credibility and significance of this category at stake. The reason for this phenomenon is, inter alia, the deficit of a thorough framework and theoretical foundations, which in turn translates into a lack of consistency in the terminology used, and in the methodology of conducting research. In line with the idea of Clark and Watson, which says that “a good theory expresses not only what a construct is, but also what it is not”, researchers dealing with the phenomenon of smartphone addiction have distinguished a definition and exclusion criteria relating to the described phenomenon. Working criteria, proposed by researchers of interdisciplinary fields, constitute a specific starting point for the development of this area of science and can be used in the diagnosis of addictions in particular, minimizing false-positive results [16,17].

### 1.2. Background

#### 1.2.1. Factors of Problematic Smartphone Use

The basis of behavioral addictions, including PMPU, is complex and unclear. The complexity results from the number of theories trying to explain the etiology of the phenomenon and the ambiguity from insufficient understanding and inability to explain its mechanisms. Currently, the following factors are listed as the underlying determinants: biogenetical, psychological, and sociocultural, which together create a biopsychosocial model of a behavioral addictions [18].

Biological and genetic factors relate primarily to the reward system and the neurotransmitter—dopamine. The physiological significance of the reward system is the secretion of dopamine as a result of taking actions of significant biological importance. It is possible to activate the reward system as a result of taking actions that are not of fundamental biological importance—for example, using social networking sites via a mobile phone or shopping leads to immediate gratification, increasing the level of dopamine and the functioning of the reward system. It should be noted that the body’s reaction to rewarding and stressful factors is an individual trait, associated with genetic and environmental determinants, as well as with the individual structure of the dopaminergic system. For example, the male sex is characterized by a higher basic activity of the discussed system, but lower reactivity to natural stimuli and psychoactive substances in relation to the females. Age is also an important issue influencing the functioning of the reward system. Puberty is a special stage during which many changes are happening in the way the brain works. The increased activation of the reward system occurring at that time and the simultaneous suppression of the systems regulating it lead to risky behaviors by young adults. Such behaviors include, inter alia, problematic use of a mobile phone or the Internet [19,20,21].

Among sociocultural factors, one can distinguish the continuous technological progress that causes a whole array of global consequences. The universal development of electronics infrastructure and simultaneous societal preoccupation with online life have both contributed to the occurrence of FOMO (fear of missing out). The idea behind this is that experiencing anxiety as a result of not being able to participate in online activities, especially those involving the current publication of content by other users. The pressure to continuously follow information on different social platforms and apps leads to an uncontrollable functioning on the Internet, hence the use of cellular phones [22,23].

#### 1.2.2. Implications of Problematic Mobile Phone Use

Consequences of the PMPU accessed via mobile phone might be divided into three categories that include somatic, mental, and social spheres of human life. The somatic sphere regards the wide range of disorders that include problems with sight and hearing, headaches and vertigo, tinnitus, carpal tunnel syndrome, and sleep disorders leading to disturbances of circadian rhythm, as well as chronic fatigue syndrome and—characteristically for smartphone users—text neck syndrome and SMS thumb due to repetitive use of handheld devices [24,25,26]. Belonging to an internet community that shows many forms of virtual pathology and generates new, artificial trends also favors the development of mental implications, which include lowered mental immunity, emotional lability, increased depressiveness, aggression, inadequate self-esteem, lowered self-value, and personality disorders [27,28].

Implications regarding the social sphere are above all related to the gradual alienation from the closest environment as a result of preoccupation with the subject of dependance. Those consequences include deterioration of social relations and loss of interpersonal skills and former interests, as well as the minimization of contact with closest ones, which lead to the sense of social isolation and loneliness [29]. On the other hand, loneliness, anxiety (including social anxiety), and depression constituting internalizing symptoms are considered not only consequences, but also important factors in the development and severity of behavioral addictions [30,31,32]. Thus, people with a high level of neuroticism characterized by an increased tendency to experience negative emotions and feelings—sadness, regret, envy, jealousy, anxiety—are more susceptible to behavioral addictions [33,34,35].

The described group of implications also refers to financial problems that are generated by the growing needs related to the growing use of a mobile phone and the desire to have more and more modern models. The problematic use of the above-mentioned technologies may also contribute to the legal consequences related to improper compliance with the law in the internet sphere. The research results also emphasize the correlation between the risky use of smartphones and the creation of dangers in the public environment, especially among pedestrians staring at the smartphone screen [36,37].

### 1.3. Aim of the Study

The aim of this study was to estimate the scale of problematic mobile phone use among young adults and to demonstrate whether negative effects such as low level of social support, increased depressiveness, and daytime sleepiness are significantly related to this phenomenon.

## 2. Materials and Methods

### 2.1. Setting and Design

The study included 567 respondents: students, graduates, and senior high school students from two cities—Koszalin and Szczecin. The study was a part of a major project. The study inclusion criteria were as follows: age above 18 and below 29 years; and providing informed consent for the participation in the study. The exclusion criteria were: lack of consent for the participation in the study as well as the age below 18 and above 29 years. Respondents received written information regarding the aim of the study as well as the possibility to withdraw consent at any stage of the study. The study was carried out in accordance with the Declaration of Helsinki and was approved by the Bioethical Commission of Pomeranian Medical University in Szczecin.

### 2.2. Research Instrument

The study was carried out using the diagnostic poll method via questionnaire technique. In the study, author’s own questionnaire form regarding sociodemographic data (sex, age, professional activity, place of residence, and marital status) as well as the following standardized research tools were used.

We used the Polish adaptation of the Mobile Phone Problem Use Scale for Adolescents (MPPUSA) by O. Lopez-Fernandez, L. Honrubia-Serrano, M. Freixa-Blanxart, and W. Gibson. The tool consists of three subscales. The first is referring to the level of mobile phone dominance over the other aspects of an individual’s life. The second one deals with the severity of preoccupation with its use, and the third one relates to the dimension of addiction and dependency on a smartphone. A respondent uses a 10-point scale where “1” stands for “not true at all” and “10” for “extremely true”. The range of results varies between 26 and 260 points and the obtained results are assigned to one of the four categories. The first category includes scores between 26 and 29 points and identifies individuals who occasionally use their mobile phones. The second group includes scores from 30 to 130 points and indicates a non-problem use of smartphones. The third category contains scores from 131 to 166 points and signalizes risk of problematic use of a mobile phone, whereas the fourth group, presence of mobile phone (dependence) problem use, is scored between 167 and 260 points [38].

We also used the Polish adaptation of The Multidimensional Scale of Perceived Social Support (MSPSS) by G. Zimet et al. The questionnaire consists of twelve statements that are related to the subjectively experienced social support by a surveyed individual from: significant others, family, and friends. The scores vary between 1 and 7 points, where results from 1 to 2.9 indicate low support, scores between 3 and 5 points indicate average support, and scores from 5.1 to 7 points are identified as high social support [39,40].

In addition, we used the Beck Depression Inventory (BDI) by A. T. Beck, R. A. Steer, and G. K. Brown in its Polish adaptation. The inventory consists of 21 positions regarding specific symptoms of depression, to which the surveyed person responds by choosing one of the four possible answers. The obtained score is a sum of points corresponding with the respondent’s answers and is assigned to one of the four following point categories: 0–11, lack of depression; 12–26, mild depression; 27–49, moderate depression; 50–63, severe depression [41,42].

The Epworth Sleepiness Scale is a questionnaire that subjectively assesses the likelihood of falling asleep during eight daily activities. The results between 0 and 9 points show regular daytime sleepiness, 10–14 indicate excessive daytime sleepiness, and scores above 14 are identified as pathological daytime sleepiness [43,44].

### 2.3. Statistical Analysis

The data gathered during the research process were collected in a Microsoft Office Excel spreadsheet. The statistical analysis was carried out using the R program, version 3.6.2. The analysis of quantitative variables was performed by calculating the mean, standard deviation, median, quartiles, and minimum and maximum. To evaluate the normality of distribution of the studied variables, we used the Shapiro–Wilk test. The Shapiro–Wilk test is a test of normality in frequentist statistics. The analysis of qualitative data was conducted by calculating the number and percentage of occurrences for each value. The comparison of qualitative variables in two groups was carried out with Mann–Whitney U test while the comparison in three or more groups was carried out with Kruskal–Wallis test, and in case of statistically significant differences, a post hoc analysis with Dunn’s test was conducted in order to identify groups that varied in a statistically significant way. The Mann–Whitney *U* test is a nonparametric
test of the null
hypothesis where, for randomly selected values *X* and *Y* from two populations, the probability of *X* being greater than *Y* is equal to the probability of *Y* being greater than *X*. The Kruskal–Wallis test is a nonparametric method for testing whether samples originate from the same distribution. It is used for comparing two or more independent samples of equal or different sample sizes. It extends the Mann–Whitney *U* test, which is used for comparing only two groups. The relationship between the analyzed parameters was evaluated using the analysis of the logistic regression model. The multivariable logistic regression was corrected for potentially distorting data (age, gender, education). The results of regression were presented with the value of the OR with 95% confidence intervals and the statistical significance value. The statistical significance level was set at 0.05; thus, all the *p* values below 0.05 were interpreted as significant [45,46].

## 3. Results

### 3.1. Characteristics of the Respondents

The study included 567 individuals, 305 of which were women (53.97%). Age of respondents varied between 18 and 29 years and was 20.53 years on average (SD = 2.78). The majority of respondents had primary education—250 of those surveyed (44.09%). Most respondents were single—311 individuals (54.85%). The largest group consisted of surveyed who lived in cities with more than 100 thousand residents—218 respondents (38.45%). The structure of professional activity showed that 351 respondents (61.90%) were unemployed and studying.

The analysis of the MPUSA regarding mobile phone use showed that 34 individuals (6%) displayed presence of mobile phone (dependence) problem use, 74 respondents (13.05%) were at risk of problematic use, while for 455 persons (80.25%) using a mobile phone was not a problem. Respondents who occasionally used their mobile phone included 4 people (0.7%).

### 3.2. Assessment of Correlation between Mobile Phone Use and Social Support, Severity of Depressiveness and Daytime Sleepiness among Respondents

The data analysis showed a statistically significant relationship between using mobile phones by the surveyed person according to the MPPUSA and their perceived social support according to the MSPSS (*p* < 0.05). It has been proven that individuals who used their phones correctly were characterized with greater support from acquaintances (*p* = 0.007 A > C) and a higher general level of support (*p* = 0.034 A > C) than ones who pathologically used their mobiles (Table 1).

The analysis of the data showed no significant relationship (*p* > 0.05) between sex, professional activity, or place of residence, as well as marital status and the prevalence of risky and problematic patterns of mobile phone use.

Data analysis also showed a significant correlation between problematic mobile phone use and being at the risk of such a thing and the severity of symptoms of depression and daytime sleepiness (*p* < 0.05). It has been established that symptoms of depression (*p* < 0.001 C,B > A) as well as excessive daytime sleepiness (*p* < 0.001 B,C > A) were much more common in respondents who presented problematic mobile phone use or were at risk of problematic use in comparison with ones who presented nonproblematic mobile phone use group. (Table 2).

Univariate and multivariate regression analysis were performed. The results are presented in Table 3. A statistically significant relationship was demonstrated between telephone abuse and an increase in the BDI score (OR = 1.032, *p* <0.001), mild depression (OR = 2.322, *p* = 0.001), and moderate depression (OR = 2.269, *p* = 0.009); and on the ESS scale (OR = 1.096, *p* <0.001), excessive somnolence (OR = 1.939, *p* = 0.005) and pathological daytime sleepiness (OR = 2.886, *p* = 0.002).

## 4. Discussion

The results of our own studies showed that the problem of mobile phone addiction concerned 6% of respondents. The mobile phone or smartphone is a modern accessory, the long-term use of which is conditioned by the multitude of available functions, which, through access to the internet, enable implementation not only in the entertainment area, but also in the professional, social, or scientific and information areas. The results of this study are consistent with the study results of other authors who also identified a minimal phenomenon of problematic mobile phone use, ranging from 1% to 7.5% [46,47,48,49].

A significantly higher percentage of people addicted to mobile phones, in relation to the result obtained in the course of this study, was obtained by authors researching Asian countries, whose cultural norms, as in the case of problematic internet use, clearly translate into increased phone use [50,51,52]. As probable causes of the discrepancies in the results, apart from sociocultural differences, we should mention methodological inconsistencies. Different tools were used in selected works. The issue of setting limit values in the standardized research tools used is also problematic. Researchers repeatedly declare a change in the limit values in the tools used, because the values adopted by the author would result in overrecognition of the studied phenomena. Therefore, it is confirmed by the fact of methodological inconsistency and it is difficult to compare different research results.

As far as the analysis of pattern of mobile phone use is concerned, no sociodemographic variable was linked with the pathologic use. However, it is worth mentioning that both women and men were addicted to using the internet and smartphone or were at risk of developing problematic patterns of use to the same degree. This result complies with the results obtained in studies by other authors. An analysis of the pattern of smartphone use among 1441 students at medical universities in China established that within the age range, phone addiction concerned 29.3% of women and 30.3% of men. The authors used a different tool than this study—the SAS-SV form—and adopted separate cut-off points for both sexes—31 points for men and 32 for women [53]. Another study included medical students from India and used the SAS-SV in order to analyze their patterns of mobile phone use with different cut-off points for both sexes. Its results showed that 45.45% of women and 57.87% of men were addicted to mobile phones [54]. Furthermore, a subjective sense of addiction was expressed by 49.23% of respondents. For reference, 34.22% of participants in this study considered themselves dependent [55]. It is also worth noting that the frequency of mobile phone use by the surveyed and their subjective sense of being addicted was much greater than the percentage of addicted individuals. Thus, long-term use of a given device should not be unequivocally identified as an addiction since, in principle, it does not only refer to the time that an individual spends using it but also to their relationship with the given device.

Our own research has shown a relationship between phone abuse and the occurrence of mild and moderate depression symptoms.

The results of this study have also demonstrated that respondents with problematic mobile phone use more frequently presented with severe depressiveness compared to the nonaddicted individuals. Other authors also analyzed the relationship between PMPU and symptoms of depression, which showed that addicted individuals much more frequently were characterized with symptoms of depression and loneliness in comparison with those who were not dependent on their phones [56,57]. An analysis of the link between smartphone addiction and depressiveness with a different research tool than in this study—the Depression Anxiety Stress Scale (DASS)—has also confirmed the presence of the described correlation. Phone-addicted individuals significantly more often showed increased depressiveness, as well as being characterized with severe anxiety and—a phenomenon specific for smartphone users—FOMO [58].

Our own research has shown a relationship between phone abuse and the occurrence of excessive sleepiness and pathological daytime sleepiness. The results of the author’s own studies identified more frequent excessive daytime sleepiness in the group of individuals who used their mobile phones pathologically. Corresponding results were obtained in two independent studies which analyzed the phonoholism phenomenon and the prevalence of sleep disorders among adolescents from Thailand and Hongkong. Authors of both studies used the same study tool as this paper—the Epworth Sleepiness Scale (ESS)—and established that individuals dependent on smartphones showed excessive daytime sleepiness more frequently than nonaddicted ones. Other authors also suggested that smartphone addiction is associated with adolescents’ poor sleep quality. It is believed that an underlying cause of sleep disorders is exposure to blue light, which is emitted by electronic devices, above all smartphones. Evening exposure to blue light reduces melatonin production—the sleep hormone that is associated with the physiological circadian rhythm—in effect causing delayed falling asleep as well as prolonged standby time [59,60,61,62,63].

Social support from parents and family but also from friends and peers is closely associated with problematic smartphone use in adolescents and young adults [64]. Other authors who researched PMPU and analyzed its relationship with perceived social support, with the same study tool as the following study, obtained similar results. A study on 1149 nursing students showed that persons who problematically used their phones had lower support from family, friends, and significant others in comparison with respondents who used their phones in proper way [65]. The study on a group of 494 Turkish students showed, however, that respondents’ PMPU was characterized with lower social support (according to the MSPSS) compared with the surveyed who were not addicted. In comparison, the author’s own studies showed that phone addicts had lower general social support as well as lower social support from acquaintances compared to non-addicts. The authors also demonstrated that individuals’ problematic mobile phone or smartphone use, besides low subjectively perceived social support, was also characterized with greater loneliness and social phobia, which translated into increased smartphone use [66,67].

## 5. Limitations

The digressions described in this study dictated certain limitations and implications for the professional practice. Problematic mobile phone use creates an important relationship between the severe symptoms of depressiveness and excessive daytime sleepiness. The pathologic pattern of phone use more frequently concerns individuals whose social support level is low. Obtained results might be useful in planning prophylactic measures and matching interventional strategies in order to minimize phone use among young adults. The main advantages of this study are research tools that were used since they were also used in the majority of studies, results of which were compared to the results of this study. An additional asset of this study is the fact that groups of both gender representatives were almost equal, which reduces the risk of biased results, which contributes to the quality and credibility of the study. The presented studies also have their disadvantages, including the number of respondents, which does not constitute a representative number for the age group that was the subject of this study. The participants filled out the questionnaire forms themselves by carrying out a self-assessment and choosing answers that resembled the reality as closely as possible. The described technique carries a risk of obtaining false answers because of conscious or subconscious answer manipulation, which increases with the level of knowledge on the subject of the study. The study did not take into account the tools that allow for the selection of people with ADHD, autism spectrum disorder, etc., among the respondents, which is another limitation of our study. The described disadvantages, however, do not disqualify the results of this study since each of the cited studies used the same methods and techniques.

## 6. Conclusions

The scale of the problematic mobile phone use phenomenon in respondents was low. Many respondents, however, expressed a subjective sense of being addicted to their smartphones in comparison with the obtained results. Problematic mobile phone use concerns members of both sexes, residents of villages and cities, as well as single people and those in relationships (both formal and informal) to the same degree. Mobile phone addiction is associated with the risk of severe symptoms of depression and excessive daytime sleepiness. Problematic mobile phone use much more frequently concerns individuals who perceive their social support level as low. The obtained results might prove useful in planning prophylactic measures and matching interventional strategies in order to minimize mobile phone and smartphone use among young adults. This study provides important findings and can provide a starting point for wider research into the concerned issues and area.

## Figures and Tables

**Table 1 ijerph-19-13549-t001:** The analysis of correlation between mobile phone use according to the MPPUSA and social support according to the MSPSS in the surveyed group.

Support		Mobile Phone Use			*p*
		Casual and Regular Users (A) N = 459	Users at Risk (B) N = 74	Problematic Users (C) N = 34	
Significant other	M ± SD	5.5 ± 1.7	5.6 ± 1.4	5.3 ± 1.5	*p* = 0.272
	Me	6.2	6	5.5	
	Q1–Q3	4.7–7	4.5–7	4.5–6.7	
Family	M ± SD	5.1 ± 1.7	4.7 ± 1.7	4.7 ± 1.7	*p* = 0.076
	Me	5.5	4.6	5	
	Q1–Q3	4–6.5	3.6–6.2	4–5.9	
Acquaintances	M ± SD	5.4 ± 1.6	5.3 ± 1.4	4.7 ± 1.6	*p* = 0.007 A > C
	Me	5.6	5.5	4.9	
	Q1–Q3	4.5–7	4.3–6.5	4–6	
General	M ± SD	5.3 ± 1.4	5.2 ± 1.2	4.9 ± 1.3	*p* = 0.034 A > C
	Me	5.6	5.3	4.9	
	Q1–Q3	4.5–6.5	4.2–6.2	4.2–5.8	

N—number of respondents, M ± SD—mean ± standard deviation, Me—median, Q1–Q3—quartiles, *p*—Kruskal–Wallis test + post hoc analysis (Dunn’s test).

**Table 2 ijerph-19-13549-t002:** The analysis of correlation between the mobile phone use according to the MPPUSA and the severity of depressiveness according to the BDI as well as daytime sleepiness according to the ESS.

Depressiveness According to the BDI	Mobile Phone Use According to the MPPUSA			*p*
	Casual and Regular Users—A (N = 459)	Users at Risk—B (N = 74)	Problematic Users—C (N = 34)	
M ± SD	11.9 ± 11.5	16.6 ± 11.7	19.5 ± 15.8	*p* < 0.001 C,B > A
Me	8	14.5	17	
Q1–Q3	3–17	7.3–23.8	5.5–32	
**Sleepiness According to the ESS**	**Casual and Regular Users—A (N = 459)**	**Users at Risk—B (N = 74)**	**Problematic Users—C (N = 34)**	** *p* **
M ± SD	8.2 ± 4.7	10.6 ± 4	10.3 ± 5.9	*p* < 0.001 B,C > A
Me	8	11	10	
Q1–Q3	5–11	8–13	5.8–14	

N—number of respondents, M ± SD—mean ± standard deviation, Me—median, Q1–Q3—quartiles, *p*—Kruskal–Wallis test + post hoc analysis (Dunn’s test).

**Table 3 ijerph-19-13549-t003:** Univariate and multivariate regression for respondents with problematic mobile phone use.

		No Adjusted				Multifactorial *			
		*p*	OR	Cl −95%	Cl +95%	*p*	OR	Cl −95%	Cl +95%
BDI	quantitative	<0.001	1.036	1.019	1.052	<0.001	1.032	1.015	1.049
	no depression		1.000				1.000		
	mild depression	0.001	2.314	1.435	3.732	0.001	2.322	1.429	3.774
	moderately severe depression	0.001	2.667	1.476	4.817	0.009	2.269	1.224	4.206
	very severe depression	0.108	4.444	0.721	27.386	0.127	4.183	0.667	26.248
ESS	quantitative	<0.001	1.106	1.057	1.157	<0.001	1.096	1.046	1.148
	normal daytime sleepiness		1.000				1.000		
	excessive daytime sleepiness	0.004	1.972	1.245	3.123	0.005	1.939	1.215	3.093
	pathological daytime sleepiness	<0.001	3.318	1.728	6.369	0.002	2.886	1.473	5.653
MPSS	quantitative	0.094	0.883	0.763	1.021	0.058	0.861	0.737	1.005
	low support		1.000				1.000		
	moderate support	0.054	2.659	0.985	7.178	0.073	2.506	0.918	6.840
	high support	0.620	1.281	0.480	3.417	0.762	1.167	0.429	3.174

Legend: OR—odds ratio, Cl—confidence interval. Notes: * Analysis adjusted by age, sex, education.

## Data Availability

Not applicable.

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
