# Peer review of "Correlations between Problematic Mobile Phone Use and Depressiveness and Daytime Sleepiness, as Well as Perceived Social Support in Adolescents"

_ijerph, 2022, doi:10.3390/ijerph192013549_

Round 1

Reviewer 1 Report

The authors present a paper on an important and timely issue. However, major revisions are required before this could could be considered for publication.

While I appreciate that the authors discuss the heterogeneity in understandings and conceptualizations of smartphone addiction, the introduction needs a lot of work. For example, it would be great if the authors could discuss some of the controversy around accepting smartphone addiction as a behavioural addiction. Moreover, there should be a more thorough discussion of previous work in this area, including the well-established links between smartphone addiction and internalizing symptoms. 

In terms of the analysis, simply looking at correlations between variables does not tell us much about these distinct groups of youth. I would recommend that the authors consider conducting a binomial logistic regression predicting membership into the "problematic" or "addicted" group. This way you could control for other known covariates (such as age and gender) and see which variables predict problematic use when considering the other factors (e.g., when we account for the influence of depression, does social support still predict problematic use?) 

A few more thoughts: 

- Be consistent in the language you are using to refer to problematic smartphone or addiction. There are currently too many different terms referencing the same construct. 

- Avoid using language like "correct" use of smartphones. It is stigmatizing and over-pathologizing. 

Author Response

28.09.2022

Dear Editors,

We take the liberty to thank you and the reviewers for insightful and careful evaluation of our article entitled “Correlations between smartphone addiction and depressiveness, daytime sleepiness as well as perceived social support in adolescents” by Marta Nowak, Kamila RachubiÅ„ska, MaÅ‚gorzata Starczewska, Ewa Kupcewicz, Aleksandra SzyliÅ„ska, Aneta Cymbaluk-PÅ‚oska and Elżbieta Grochans and for allowing us to resubmit a revised manuscript.

The comments helped us to improve the quality of the manuscript. We considered all comments and recommendations and responded to Reviewers’ questions. The corrections throughout the manuscript were marked in blue.

Our responses to the reviews are attached below.

Thank you for your consideration. We look forward to hearing from you.

Sincerely,

Kamila Rachubińska

Open Review

English language and style

(x) Extensive editing of English language and style required
( ) Moderate English changes required
( ) English language and style are fine/minor spell check required
( ) I don't feel qualified to judge about the English language and style

Yes

Can be improved

Must be improved

Not applicable

Does the introduction provide sufficient background and include all relevant references?

()

( )

(x)

( )

Are all the cited references relevant to the research?

( )

(x)

( )

( )

Is the research design appropriate?

()

( )

(x)

( )

Are the methods adequately described?

()

(x)

( )

( )

Are the results clearly presented?

()

( )

(x)

( )

Are the conclusions supported by the results?

()

(x)

( )

( )

Comments and Suggestions for Authors

The authors present a paper on an important and timely issue. However, major revisions are required before this could be considered for publication.

While I appreciate that the authors discuss the heterogeneity in understandings and conceptualizations of smartphone addiction, the introduction needs a lot of work. For example, it would be great if the authors could discuss some of the controversy around accepting smartphone addiction as a behavioral addiction. Moreover, there should be a more thorough discussion of previous work in this area, including the well-established links between smartphone addiction and internalizing symptoms. 

RESPONSE: Thank you for your review. Thank you very much for a thorough editorial assessment of my manuscript, positive opinions, as well as the reviewers’ remarks. I used them as an important guide to improving the quality of my paper. The implemented corrections were done strictly according to their comments. I have enclosed the re-edited manuscript and cover letter as responses to Reviewers, detailing how I followed their suggestions. We have added a description of the controversy surrounding the acceptance of smartphone addiction as a behavioral addiction. We've covered smartphone addiction in more detail.

In terms of the analysis, simply looking at correlations between variables does not tell us much about these distinct groups of youth. I would recommend that the authors consider conducting a binomial logistic regression predicting membership into the "problematic" or "addicted" group. This way you could control for other known covariates (such as age and gender) and see which variables predict problematic use when considering the other factors (e.g., when we account for the influence of depression, does social support still predict problematic use?) 

RESPONSE: Thank you for this suggestion. We conducted binomial logistic regression as shown in Results section Table 3.

A few more thoughts: 

- Be consistent in the language you are using to refer to problematic smartphone or addiction. There are currently too many different terms referencing the same construct. 

- Avoid using language like "correct" use of smartphones. It is stigmatizing and over-pathologizing. 

RESPONSE: Thank you for your comments. We checked and corrected all mistakes.

Yours faithfully,

Kamila Rachubińska

Kamila Rachubińska, corresponding author

Department of Nursing, Faculty of Health Sciences, Pomeranian Medical University in Szczecin,
Head: Prof. Elżbieta Grochans

48 Å»oÅ‚nierska St., 71 – 210 Szczecin, Poland

Tel. (091) 48-00-910

E-mail: kamila.rachubinska@pum.edu.pl

Reviewer 2 Report

I have read this very interesting report on an important growing area of concern in the addiction sciences related to excessive smartphone use.  While I consider this topic to be of importance and I commend the authors for engaging in this study, my enthusiasm was dampened by 3 major issues. First is grammar and language.  I can appreciate that English is not the native language of the authors. However, the use of incorrect nouns, confusing sentence structure and other grammatical errors rendered the report very difficult to understand.  Second is the lack of statistical details in the methods and third is the organization of the Discussion.  I have detailed these issues and others below.   

1. line 21.  It is not clear what is meant by "in comparison with the ones who used them in a correct way."  What is correct vs. incorrect phone use?   Perhaps rephrasing this to "individuals who used their phones less frequently"?  I understood after reading the Methods what "correct" use is a term defined by the survey but this is not at all clear from the abstract.

2. Why is "Internet" capitalized throughout? It is not a proper name.

3. In the Results section, it is not clear which statistical approaches were used to generate each of the p values.  Please indicate the full statistics, including value of the r values for correlations, Chi square value, U values etc.

4. From my read of the Methods, childhood mental or health disorders were not considered for exclusion criterion.  Did the authors find any relationship between a pre-existing (or early diagnosis) condition and problematic cell phone use (e.g., ADHD, depression, autism spectrum etc)?

5.  Please compile the very small paragraphs related to findings from individual studies listed across Pages 5-7 into a more comprehensive paragraph or set of larger paragraphs with a clear take-home message.  As currently written, this section of the Discussion reads like a list  instead of an integrated discussion of past findings in relation to present findings.

6. In re-reading the Introduction, this section also suffers from short, incomplete paragraphs that would be combined (e.g., paragraphs on phantom vibration and phantom ringtone).

7. As this report might generate interest from scientists (such as myself) who study substance use disorders but not behavioral addictions, some sample questions from the cell phone surveys would provide insight into the types of questions asked of the subjects.  Other surveys like the Beckman Depression Inventory are more common and examples need not be provided.

8.  There are too many grammatical issues for me to highlight them all.  Here are some grammatical errors early in the report.

First sentence: "The definition of cell addiction, similarly to the other behavioral addictions, is heterogenous and depends on an author’s approach".  Problem 1:  The term "cell addiction" is never used again in the report and so one is left to wonder if it's the same as mobile phone addiction (which is more commonly used in this report).  Problem 2: "similarly to..." should read "similar to". Problem 3:  "depends on an author's approach".  This is not clarified at all.  As written, it sounds like every researcher makes up their own definition of mobile phone addiction and that there is no standard as there is for gambling or eating disorders. If that is the case, then it should be stated as such but should be explained that individuals use different measures or scales to diagnose.

line 46:  "Background of behavioral addictions...".  I believe the authors are referring to the "cause" of behavioral addictions.  A disease state does not usually have a "background".  Individual people have a background.  This is just one of many, many examples where the word choice is incorrect.

Author Response

28.09.2022

Dear Editors,

We take the liberty to thank you and the reviewers for insightful and careful evaluation of our article entitled “Correlations between smartphone addiction and depressiveness, daytime sleepiness as well as perceived social support in adolescents” by Marta Nowak, Kamila RachubiÅ„ska, MaÅ‚gorzata Starczewska, Ewa Kupcewicz, Aleksandra SzyliÅ„ska, Aneta Cymbaluk-PÅ‚oska and Elżbieta Grochans and for allowing us to resubmit a revised manuscript.

The comments helped us to improve the quality of the manuscript. We considered all comments and recommendations and responded to Reviewers’ questions. The corrections throughout the manuscript were marked in blue.

Our responses to the reviews are attached below.

Thank you for your consideration. We look forward to hearing from you.

Sincerely,

Kamila Rachubińska

Open Review

() I would not like to sign my review report
(x ) I would like to sign my review report

English language and style

(x) Extensive editing of English language and style required
( ) Moderate English changes required
( ) English language and style are fine/minor spell check required
( ) I don't feel qualified to judge about the English language and style

Yes

Can be improved

Must be improved

Not applicable

Does the introduction provide sufficient background and include all relevant references?

( )

(x)

( )

( )

Are all the cited references relevant to the research?

(x)

( )

( )

( )

Is the research design appropriate?

(x)

( )

( )

( )

Are the methods adequately described?

()

(x)

( )

( )

Are the results clearly presented?

()

( )

(x)

( )

Are the conclusions supported by the results?

()

(x)

( )

( )

Comments and Suggestions for Authors

I have read this very interesting report on an important growing area of concern in the addiction sciences related to excessive smartphone use.  While I consider this topic to be of importance and I commend the authors for engaging in this study, my enthusiasm was dampened by 3 major issues. First is grammar and language.  I can appreciate that English is not the native language of the authors. However, the use of incorrect nouns, confusing sentence structure and other grammatical errors rendered the report very difficult to understand.  Second is the lack of statistical details in the methods and third is the organization of the Discussion.  I have detailed these issues and others below.   

RESPONSE: Thank you very much for a thorough editorial assessment of my manuscript, positive opinions, as well as the reviewers’ remarks. I used them as an important guide to improving the quality of my paper. The implemented corrections were done strictly according to their comments. I have enclosed the re-edited manuscript and cover letter as responses to Reviewers, detailing how I followed their suggestions.    

  1. line 21.  It is not clear what is meant by "in comparison with the ones who used them in a correct way."  What is correct vs. incorrect phone use?   Perhaps rephrasing this to "individuals who used their phones less frequently"?  I understood after reading the Methods what "correct" use is a term defined by the survey but this is not at all clear from the abstract.

RESPONSE: Thank you for this suggestion. We have changed it.

  1. Why is "Internet" capitalized throughout? It is not a proper name.

RESPONSE: Thank you for this comment. We have changed it.

  1. In the Results section, it is not clear which statistical approaches were used to generate each of the p values.  Please indicate the full statistics, including value of the r values for correlations, Chi square value, U values etc.

RESPONSE: Thank you for this comments, we have added the missing data.

  1. From my read of the Methods, childhood mental or health disorders were not considered for exclusion criterion.  Did the authors find any relationship between a pre-existing (or early diagnosis) condition and problematic cell phone use (e.g., ADHD, depression, autism spectrum etc)?

RESPONSE: Thank you for this comment. The study did not take into account the tools enabling the selection of people with ADHD, autism spectrum, etc. among the respondents, which is one of the limitations of our study. This limitation is described in the Limitations section. Thank you for your valuable comment, we will use the clue to conduct further research in this direction.

  1. Please compile the very small paragraphs related to findings from individual studies listed across Pages 5-7 into a more comprehensive paragraph or set of larger paragraphs with a clear take-home message.  As currently written, this section of the Discussion reads like a list  instead of an integrated discussion of past findings in relation to present findings.

RESPONSE: Thank you for this comment, we have added the missing data.

  1. In re-reading the Introduction, this section also suffers from short, incomplete paragraphs that would be combined (e.g., paragraphs on phantom vibration and phantom ringtone).

RESPONSE: Thank you for this suggestion. We have changed it.

  1. As this report might generate interest from scientists (such as myself) who study substance use disorders but not behavioral addictions, some sample questions from the cell phone surveys would provide insight into the types of questions asked of the subjects.  Other surveys like the Beckman Depression Inventory are more common and examples need not be provided.

RESPONSE: Thank you for this comment. We will use these types of questions while conducting further research.

  1. There are too many grammatical issues for me to highlight them all.  Here are some grammatical errors early in the report.

First sentence: "The definition of cell addiction, similarly to the other behavioral addictions, is heterogenous and depends on an author’s approach".  Problem 1:  The term "cell addiction" is never used again in the report and so one is left to wonder if it's the same as mobile phone addiction (which is more commonly used in this report).  Problem 2: "similarly to..." should read "similar to". Problem 3:  "depends on an author's approach".  This is not clarified at all.  As written, it sounds like every researcher makes up their own definition of mobile phone addiction and that there is no standard as there is for gambling or eating disorders. If that is the case, then it should be stated as such but should be explained that individuals use different measures or scales to diagnose.

line 46:  "Background of behavioral addictions...".  I believe the authors are referring to the "cause" of behavioral addictions.  A disease state does not usually have a "background".  Individual people have a background.  This is just one of many, many examples where the word choice is incorrect.

RESPONSE: Thank you for these comments, we checked and corrected all mistakes.

Yours faithfully,

Kamila Rachubińska

Kamila Rachubińska, corresponding author

Department of Nursing, Faculty of Health Sciences, Pomeranian Medical University in Szczecin,
Head: Prof. Elżbieta Grochans

48 Å»oÅ‚nierska St., 71 – 210 Szczecin, Poland

Tel. (091) 48-00-910

E-mail: kamila.rachubinska@pum.edu.pl

Reviewer 3 Report

The paper “Correlations between smartphone addiction and depressiveness, daytime
sleepiness as well as perceived social support in adolescents” presents interesting analyzes to estimate the scale of mobile phone addiction among young adults as well as to establish whether the low level of perceived social support is related to the problematic smartphone use, and whether an addictive pattern of smartphone use is related to the prevalence of depressiveness and excessive daytime sleepiness. However, the manuscript must be improved in the following aspects:

  1. Introduction: Include a paragraph presenting the structure of the paper: “This paper is organized in X Sections. Section 1 presents...”

  2. Include a Section of Background or Theoretical Foundation.

  3. Statistical analysis: more information about the statistical analysis should be presented.

  4. It is suggested to present information on equations and statistical tests adopted. In addition, provide the appropriate references for the tests used.

  5. Conclusions: include proposals for future work.

  6. References: include more recent references: last three years (2020, 2021 and 2022)

Author Response

28.09.2022

Dear Editors,

We take the liberty to thank you and the reviewers for insightful and careful evaluation of our article entitled “Correlations between smartphone addiction and depressiveness, daytime sleepiness as well as perceived social support in adolescents” by Marta Nowak, Kamila RachubiÅ„ska, MaÅ‚gorzata Starczewska, Ewa Kupcewicz, Aleksandra SzyliÅ„ska, Aneta Cymbaluk-PÅ‚oska and Elżbieta Grochans and for allowing us to resubmit a revised manuscript.

The comments helped us to improve the quality of the manuscript. We considered all comments and recommendations and responded to Reviewers’ questions. The corrections throughout the manuscript were marked in blue.

Our responses to the reviews are attached below.

Thank you for your consideration. We look forward to hearing from you.

Sincerely,

Kamila Rachubińska

Open Review

English language and style

() Extensive editing of English language and style required
(x ) Moderate English changes required
( ) English language and style are fine/minor spell check required
( ) I don't feel qualified to judge about the English language and style

Yes

Can be improved

Must be improved

Not applicable

Does the introduction provide sufficient background and include all relevant references?

( )

(x)

( )

( )

Are all the cited references relevant to the research?

( )

(x)

( )

( )

Is the research design appropriate?

( )

(x)

( )

( )

Are the methods adequately described?

()

(x)

( )

( )

Are the results clearly presented?

()

(x)

()

( )

Are the conclusions supported by the results?

()

(x)

( )

( )

Comments and Suggestions for Authors

The paper “Correlations between smartphone addiction and depressiveness, daytime
sleepiness as well as perceived social support in adolescents” presents interesting analyzes to estimate the scale of mobile phone addiction among young adults as well as to establish whether the low level of perceived social support is related to the problematic smartphone use, and whether an addictive pattern of smartphone use is related to the prevalence of depressiveness and excessive daytime sleepiness. However, the manuscript must be improved in the following aspects:

  1. Introduction: Include a paragraph presenting the structure of the paper: “This paper is organized in X Sections. Section 1 presents...”

RESPONSE: Thank you for your review. We have added a paragraph presenting the structure of the paper.

  1. Include a Section of Background or Theoretical Foundation.

RESPONSE: Thank you for this comment. We have added a Section of Background.

  1. Statistical analysis: more information about the statistical analysis should be presented.

RESPONSE: Thank you for this suggestion. The missing data have been added.

  1. It is suggested to present information on equations and statistical tests adopted. In addition, provide the appropriate references for the tests used.

RESPONSE: Thank you for this suggestion

  1. Conclusions: include proposals for future work.

RESPONSE: Thank you for this comment. We have added proposals for future work to conclusions section.

  1. References: include more recent references: last three years (2020, 2021 and 2022)

RESPONSE: Thank you for this comment. We have reviewed and added more recent references.

Yours faithfully,

Kamila Rachubińska

Kamila Rachubińska, corresponding author

Department of Nursing, Faculty of Health Sciences, Pomeranian Medical University in Szczecin,
Head: Prof. Elżbieta Grochans

48 Å»oÅ‚nierska St., 71 – 210 Szczecin, Poland

Tel. (091) 48-00-910

E-mail: kamila.rachubinska@pum.edu.pl

Round 2

Reviewer 2 Report

I have read the revised manuscript and the authors have worked hard to address all of my prior concerns.  I have no further comments at the present time.

Author Response

10.10.2022

Dear Editors,

We take the liberty to thank you and the reviewers for insightful and careful evaluation of our article entitled “Correlations between smartphone addiction and depressiveness, daytime sleepiness as well as perceived social support in adolescents” by Marta Nowak, Kamila RachubiÅ„ska, MaÅ‚gorzata Starczewska, Ewa Kupcewicz, Aleksandra SzyliÅ„ska, Aneta Cymbaluk-PÅ‚oska and Elżbieta Grochans and for allowing us to resubmit a revised manuscript.

The comments helped us to improve the quality of the manuscript. We considered all comments and recommendations and responded to Reviewers’ questions. Our responses to the reviews are attached below.

Thank you for your consideration. We look forward to hearing from you.

Sincerely,

Kamila Rachubińska

Open Review

() I would not like to sign my review report
(x ) I would like to sign my review report

English language and style

( ) Extensive editing of English language and style required
( ) Moderate English changes required
(x) English language and style are fine/minor spell check required
( ) I don't feel qualified to judge about the English language and style

Yes

Can be improved

Must be improved

Not applicable

Does the introduction provide sufficient background and include all relevant references?

(x)

( )

( )

( )

Are all the cited references relevant to the research?

(x)

( )

( )

( )

Is the research design appropriate?

(x)

( )

( )

( )

Are the methods adequately described?

(x)

( )

( )

( )

Are the results clearly presented?

(x)

( )

( )

( )

Are the conclusions supported by the results?

(x)

( )

( )

( )

Comments and Suggestions for Authors

I have read the revised manuscript and the authors have worked hard to address all of my prior concerns.  I have no further comments at the present time.

RESPONSE: Thank you very much for an editorial assessment of my manuscript, positive opinions, as well as the reviewers’ remarks. I used them as an important guide to improving the quality of my paper.

Yours faithfully,

Kamila Rachubińska

Kamila Rachubińska, corresponding author

Department of Nursing, Faculty of Health Sciences, Pomeranian Medical University in Szczecin,
Head: Prof. Elżbieta Grochans

48 Å»oÅ‚nierska St., 71 – 210 Szczecin, Poland

Tel. (091) 48-00-910

E-mail: kamila.rachubinska@pum.edu.pl

Reviewer 3 Report

The authors made changes to the paper. However, the work still lacks improvements in the structure. Thus, it is again suggested that:

1) Introduction: The Introduction section is still not well structured. I don't find interesting subsections in the introduction. 

2) Section of Background or Theoretical Foundation should be created after the Introduction.

3) Statistical analysis: more information about the statistical analysis should be presented in Section 2.3. It is suggested to present more information on equations and statistical tests adopted. 

Author Response

10.10.2022

Dear Editors,

We take the liberty to thank you and the reviewers for insightful and careful evaluation of our article entitled “Correlations between smartphone addiction and depressiveness, daytime sleepiness as well as perceived social support in adolescents” by Marta Nowak, Kamila RachubiÅ„ska, MaÅ‚gorzata Starczewska, Ewa Kupcewicz, Aleksanda SzyliÅ„ska, Aneta Cymbaluk-PÅ‚oska and Elżbieta Grochans and for allowing us to resubmit a revised manuscript.

The comments helped us to improve the quality of the manuscript. We considered all comments and recommendations and responded to Reviewers’ questions. The correction throughout the manuscript were marked in yellow.

Our responses to the reviews are attached below.

Thank you for your consideration. We look forward to hearing from you.

Sincerely,

Kamila Rachubińska

Open Review

English language and style

( ) Extensive editing of English language and style required
( ) Moderate English changes required
(x) English language and style are fine/minor spell check required
( ) I don't feel qualified to judge about the English language and style

Yes

Can be improved

Must be improved

Not applicable

Does the introduction provide sufficient background and include all relevant references?

( )

( )

(x)

( )

Are all the cited references relevant to the research?

(x)

( )

( )

( )

Is the research design appropriate?

( )

(x)

( )

( )

Are the methods adequately described?

()

( )

(x)

( )

Are the results clearly presented?

()

(x)

()

( )

Are the conclusions supported by the results?

(x)

( )

( )

( )

Comments and Suggestions for Authors

The authors made changes to the paper. However, the work still lacks improvements in the structure. Thus, it is again suggested that:

  1. Introduction: The Introduction section is still not well structured. I don't find interesting subsections in the introduction. 

RESPONSE: Thank you for this suggestion. The missing data have been added.

  1. Section of Background or Theoretical Foundation should be created after the Introduction.

RESPONSE: Thank you for this comments. We changed it.

  1.  Statistical analysis: more information about the statistical analysis should be presented in Section 2.3. It is suggested to present more information on equations and statistical tests adopted.

RESPONSE: Thank you for this suggestion. The missing data have been added. 

Yours faithfully,

Kamila Rachubińska

Kamila Rachubińska, corresponding author

Department of Nursing, Faculty of Health Sciences, Pomeranian Medical University in Szczecin,
Head: Prof. Elżbieta Grochans

48 Å»oÅ‚nierska St., 71 – 210 Szczecin, Poland

Tel. (091) 48-00-910

E-mail: kamila.rachubinska@pum.edu.pl
